# Probiotic as a Potential Gut Microbiome Modifier for Stroke Treatment: A Systematic Scoping Review of In Vitro and In Vivo Studies

**DOI:** 10.3390/nu14173661

**Published:** 2022-09-05

**Authors:** Chatuthanai Savigamin, Chatpol Samuthpongtorn, Nuttida Mahakit, Tanawin Nopsopon, Julia Heath, Krit Pongpirul

**Affiliations:** 1Department of Parasitology, Chulalongkorn University Faculty of Medicine, Bangkok 10330, Thailand; 2Department of Preventive and Social Medicine, School of Global Health, Chulalongkorn University Faculty of Medicine, Bangkok 10330, Thailand; 3Department of Epidemiology, Harvard T.H. Chan School of Public Health, Boston, MA 02115, USA; 4Department of Medicine, Brigham and Women’s Hospital, Boston, MA 02115, USA; 5Department of International Health, Johns Hopkins Bloomberg School of Public Health, Baltimore, MD 21205, USA; 6Clinical Research Center, Bumrungrad International Hospital, Bangkok 10110, Thailand

**Keywords:** probiotic, stroke, gut microbiome, meta-analysis

## Abstract

Background: Pharmacologic and non-pharmacologic treatments for stroke are essential but can be costly or harmful, whereas probiotics are a promising alternative. This scoping review aimed to synthesize the *in vitro* and *in vivo* evidence of probiotics on stroke-related neurological, biochemical, and histochemical outcomes. Methods: A systematic review was conducted in PubMed, Embase, and Cochrane Central Register of Clinical Trials (CENTRAL) up to 7 May 2021. Titles and abstracts were screened and assessed by two independent reviewers. The initial screening criteria aimed to include studies using probiotics, prebiotics, and symbiotics both *in vitro* and *in vivo* for the prevention and/or treatment of stroke. Results: Of 6293 articles, 4990 passed the initial screen after excluding duplication articles, of which 36 theme-related full texts were assessed and 13 were included in this review. No *in vitro* studies passed the criteria to be included in this review. Probiotics can ameliorate neurological deficits and show their anti-inflammation and anti-oxidative properties. Decreased loss of cerebral volume and inhibition of neuronal apoptosis were revealed in histopathological evidence. Conclusions: There are potential cognitive benefits of probiotic supplementation, especially among animal models, on decreasing cerebral volume, increasing neurological score, and decreasing the inflammatory response. However, further investigation is needed to validate these conclusions in various populations.

## 1. Introduction

Stroke is a cerebrovascular disease that has been known to be a significant burden in low- to middle-income countries and to be declining in high-income countries. In 2019, stroke was identified as the second leading cause of disability-adjusted life years (DALYs) worldwide [1]. Numerous rehabilitation programs and interventions that require multimodality teams, both pharmacologic and nonpharmacologic, have been conducted. However, the outcomes have not met the needs of the burden of this disease [2]. Several monoaminergic drugs have been investigated for use in stroke, including serotonergic and dopaminergic drugs [3], but these drugs may cause a variety of side effects, including sexual dysfunction, weight gain, insomnia, or somnolence in selective serotonin reuptake inhibitors (SSRIs), and gastrointestinal side effects, motor disturbances, and others in dopaminergic drugs [4,5]. These potential side effects may have a significant impact on the patient’s quality of life if taken for an extended period. Stroke patients require interventions with lower costs and fewer side effects, such as probiotics. Probiotics are live microorganisms found in a variety of foods that may benefit a variety of human organ systems and conditions, including the immune system, dysbiosis, and others [6]. A recent study discovered that the gut microbiota–brain axis, which is involved in multiple systems, including the endocrine, neurological, and digestive systems, had a direct effect on these systems and may be used cautiously in certain psychiatric diseases such as depression [7]. The gut microbiota–brain axis was also discussed in a study involving stroke and the microbiota, in which it was found that altering the gut microbiota could affect stroke outcome by impacting both neuroendocrine and metabolic pathways including the serotoninergic system, γ-aminobutyric acid (GABA), plasma cortisol and catecholamine [8,9].

Moreover, previous data found that microbiota could impact the central nervous system by using the vagus nerve [10] and local signals including histamine, acetylcholine, melatonin, nitric oxide and hydrogen oxide, which can affect enteroendocrine cells. The microbiome has been known to play an essential part in microglial maturation and activation which can change behavior in studied samples [11]. Furthermore, mice given different foods present different physical activity, memory and anxiety behavior, which shows the impact microbiota have via bottom-up pathways [9,12,13]. 

These pathways are further investigated in other studies looking at the use of probiotics to alter the gut microbiota–brain axis, which found that probiotic use could improve the outcome of stroke model disease [14]. To our knowledge, few systematic reviews have been conducted on this topic and additional investigation is required. Therefore, the aim of this systematic review is to further investigate the outcome of probiotic use for the treatment of *in vitro* and *in vivo* study models of stroke disease and see if it could serve as part of an effective, low-cost treatment plan in some stroke patients.

## 2. Materials and Methods

### 2.1. Registration of Protocol

This study was guided by the recommendation of the Preferred Reporting Items of Systematic reviews and Meta-Analyses extension for Scoping Review (PRISMA-ScR) statement. 

### 2.2. Data Sources and Search Strategy

We used three databases including PubMed, Embase, and Cochrane Central Register of Clinical Trials (CENTRAL) due to their coverage of international peer review articles to search for publications in the English language up to 7 May 2021. The terms probiotics, prebiotics, and symbiotics were used in combination with ischemic stroke, cerebrovascular disorders, and brain ischemia as the keywords for a systematic literature search along with any synonyms. The details of the search terms are presented in the Appendix A. In addition, the reference lists of included articles were searched, as well as relevant citations from other journals via Google Scholar.

### 2.3. Study Selection

In this systematic scoping review, we worked with an information specialist to design an appropriate search strategy to identify original peer-reviewed articles that look at the use of probiotics, symbiotics, or prebiotics as a form of prevention or treatment for stroke. We defined that the population of each study could be conducted either in vitro (using components of an organism that have been isolated from their usual biological surroundings, ex. microorganisms, cells, or biological molecules) or in vivo (living organisms, ex. animals). The interventions were probiotics, symbiotics, or prebiotics compared with placebo. Probiotics were further categorized into food and non-food based. Food-based probiotics ranged from ginseng and yam gruel to fermented soybeans, which have been traditionally used in Chinese traditional medicine [15]. Non-food based probiotics were composed of a variety of probiotics ranging from single-strain bacteria, such as *Clostridium butyricum* or lactobacillus, to a mixture of probiotics. *Clostridium butyricum* has been known to be a probiotic commonly used and studied in Asia with an effect on modulating immune processes and inflammatory processes in the intestinal tract [16]. Article screening for eligible studies that correlated with our inclusion and exclusion criteria was conducted by two independent reviewers (C.Sam. and T.N.). The independent reviewers were selected for their experience in probiotics, stroke, and systematic review methodology. Discrepancies between the two reviewers were resolved by consensus.

### 2.4. Data Extraction

Data extraction was done by two independent reviewers (C.S. and C.Sam.) for a published summary of probiotic effects in stroke animals and in vitro studies. The following data were extracted: study characteristics (authors, year of publication, study type, journal name, contact information, country, and funding); intervention characteristics (probiotics, symbiotics, and prebiotics; food-based vs non-food based; the number of case and control); and outcome characteristics (neurological outcome, biochemical profile, dysbiosis index, and histopathology). We included all associated text, tables, and figures for extraction. We excluded non-original articles (review articles, protocols, letters, comments, and guidelines); non-human studies; unpublished data or non-peer-reviewed data; and studies published in languages other than English.

### 2.5. Data Synthesis and Analysis

The primary outcomes included in the systematic review are neurological outcome, biochemical profile, dysbiosis index, and histopathology among *in vitro* and *in vivo* study models. 

### 2.6. Patient and Public Involvement

There was no patient or public involvement in the study. However, the results of the studies included in this review (both *in vivo* and *in vitro*) could promote further research on this topic.

## 3. Results

### 3.1. Study Characteristics

The database search identified 6293 potential records. After removing duplicates, 4990 titles passed the initial screen, and 36 theme-related abstracts were selected as further full-text articles to be assessed for eligibility (Figure 1). A total of 23 articles were excluded for the following: 6 non-peer-reviewed, 4 duplicates, 4 protocols, 4 wrong interventions using something other than probiotics/prebiotics and symbiotics as an intervention, 3 wrong outcomes i.e., using outcomes other than neurological outcomes such as gastrointestinal symptoms, and 2 non-English languages. Thirteen studies were eligible for the data extraction and data synthesis.

The 13 included studies investigated probiotic interventions, neurological outcomes, biochemical changes, and/or histopathological findings. All studies were animal studies with eight rats (61.5%), four mice (30.8%), and one gerbil (7.7%) as a study model. The study period was from 2004 to 2020. To the best of our knowledge, *in vitro* studies that correlate with our criteria in all of the databases were not found. The studies took place in Northern America, Europe, and Asia. There were no studies from Australia, Africa, and Southern America.

### 3.2. Summary of Probiotic Impact in Stroke-Induced Animal

There were multiple parameters used to evaluate the effect of probiotics in the animal model. Of the 13 included articles, both neurological outcomes and biochemical profiles were measured in 11 articles (84.6%). Similarly, histopathology was measured in 11 out of 13 studies (84.6%) (Table 1).

#### 3.2.1. Neurological Test

Most of the studies [14,18,19,21,23,24,25,26,27,28] showed that probiotic use could decrease neurological deficits and some claimed that it can improve spatial learning ability and cognitive function. Tests used to measure cognitive functioning in the studies include neurological function score [14,18,19,20,21,22,23,24,25,26,27,28,29], Morris water maze [20,22,23], and open field test [20]. Morris water mazes were used to evaluate the capability of rodent spatial learning by navigating and locating the escape platform of the maze [30]. Open field tests were used in the observation of exploratory behavior by focusing on the movement activity of rodents, which could be related to motor power, emotional, and other instinctive behaviors such as fight and fear [31].

#### 3.2.2. Biochemical Level

The inflammatory response was also found to be impacted by probiotic use, including Tumor necrosis factor (TNF) alpha [32,33] and Interleukin 1 (IL-1) [32,33], which show an anti-inflammatory effect with probiotics [21,23]. TNF alpha is secreted by macrophages and is responsible for acute inflammation processes that lead to cell necrosis or apoptosis [32]. IL-1 is known to be the regulator of inflammation and is involved in a variety of pathways, including the innate immune process which causes a range of functions from leukocytic pyrogen and fever to an activated immune system [34]. Other studies found that probiotics are involved in decreasing multiple factors in metabolic pathways such as SCFAs, lipid profile and blood sugar. SCFAs have been studied extensively in gut-associated effects including stabilized intestinal membrane integrity, anti-inflammation and production of mucous. Moreover, SCFAs were found to cross the blood–brain barrier and control the blood–brain barrier’s integrity which impacts the gut–brain axis [35,36]. Similarly, hyperglycemia has been found to cause a larger infarction area in a mouse model which may be due to its toxicity, but the mechanism is still not clear [37].

Another dimension was associated with that involved in the coagulation pathway, which could impact the resolution of stroke in animal models. A high fibrinogen level was associated with an increased risk of mortality and predicted poor outcome in an ischemic stroke model [38,39]. Decreased oxidative stress has also been reported after using probiotics [15,21] by using serum malondialdehyde (MDA), which is a product in lipid peroxidation that can be inferred as a biomarker for oxidative stress. Superoxide dismutase (SOD) is an antioxidant that has a major role in preventing both endothelial and mitochondrial dysfunction. SOD impacts ischemic penumbra by reducing superoxide anion and abnormal glutamate release [40,41,42].

Nuclear factor erythroid 2-related factor 2 (Nrf2) [28] was one of the novel targets in the treatment of stroke due to its ability to induce the antioxidative potential of the cell especially in neurons by cooperating with its actin-binding protein Keap1 [43,44]. Brain-derived neurotrophic factor (BDNF) level—one of the important factors involved in neuron differentiation and survival [45]—was used for measuring the outcome in two studies [19,25].

#### 3.2.3. Histopathology

Some researchers have found that probiotics could be used to decrease cerebral infarction volume [27,28,30,32,46,47]. Using cerebral infarction volume is known as one of the stronger predictors in acute stoke outcomes from previous multivariate analysis, which suggests the reliability of using this outcome as a measurement in almost all of the study that had been included [48]. Moreover, probiotics have been found to inhibit neuronal apoptosis by detecting the expression of caspase-3, Akt, Iba1, TLR4, IkB and A20 [46,49]. Caspase-3 is known as a key factor in neuron cell death within the acute phase of stroke and could be used as a interesting marker of cell apoptosis [48]. Probiotics can also play a role in the immunologic pathway of the brain and decrease changes within the hippocampus [34,50].

## 4. Discussion

This systematic review showed that a variety of probiotic types have been used in interventions for potential stroke treatment, which makes it difficult to compare results between the studies. Despite these differences, we found similar results in multiple parameters including neurological score and brain volume, which suggests that probiotic supplementation could be a beneficial component of rehabilitation programs in post-stroke patients. For example, a systematic review of probiotic use in human stroke patients in other modalities was also published and found that probiotics in enteral nutrition can improve clinical parameters including infection events, intestinal dysbiosis, gastrointestinal complications, and nutritional status [30]. However, more research is needed due to the lack of literature available on probiotic supplementation in human specimens and its effect on neurological modalities.

### 4.1. Probiotic and Inflammatory Effects

There is growing evidence that supports beneficial effects of inflammatory processes in various aspects of ischemic cerebral disease. To illustrate a bidirectional process that involved peripheral immune response and the cerebral ischemic brain model, many pathways were found in the inflammatory process including: interleukin 1 as protection of hippocampal cortical area 1 (CA1) in gerbil, T-reg in postischemic anti-inflammatory effect through use of IL-10 secretion in ischemic brain tissue, and the complex role of neutrophil that could destabilize the blood–brain barrier by enzyme secretion and create inflammation by blocking vascular supply [31]. These pathways are linked to the essential role of inflammation in ischemic brain disease.

Although not fully understood, the gut–brain axis has been known to be the novel bidirectional pathway involving the gut and brain. Moreover, it is thought to involve the vagus nerve in multiple complex pathways including the enteric nervous system, neuronal-glial endothelial interaction, and immune and inflammatory cell response involving damage-associated molecular pattern (DAMPS) and cytokine. Previous associations have also been found to suggest gut dysbiosis or an altered strain of gut microbiota to be preset in stroke patients [32]. Thus, some studies have included probiotic supplementation as a primary form of intervention and found probiotic use to improve neurological outcomes by using bidirectional pathways [33].

### 4.2. The Use of the Hippocampus in Measured Outcomes

The hippocampus is a known part of the limbic system and plays a critical role in many neurological diseases, including Alzheimer’s disease. This part of the brain has various functions, including memory, mood, emotional drive, and spatial navigation. Thus, much of the research presented in our study used many components from the hippocampus, such as neuron loss as the neurological outcome, to evaluate the effect of probiotics in stroke animal models [34].

### 4.3. Probiotic Strain and Its Benefit

In this study we found that using different types of probiotic could impact the outcome of the disease, for example Panax Notoginsenoside Extract (PNE) was found to affect the BDNF expression [19]. On the other hand, commercial probiotics (LactoCare capsule) did not have the same impact [25] which shows the importance of selection of strain and type of probiotic that could be further studied in human or animal models. This is similar to a previous review article which showed the potential benefit of probiotics in various diseases according to the strain, from gastrointestinal-related diseases including colitis, inflammatory bowel disease, irritable bowel syndrome and diarrhea, to non-gastrointestinal diseases such as obesity and Alzheimer’s disease.

Several probiotics with anti-inflammation effects that could be further investigated in stroke-associated models include *Lactiplantibacillus plantarum, Lacticaseibacillus rhamnosusc* and *Bifidobacterium infantis Bifidobacterium breve*, which have been previously shown to have benefits in spontaneous colitis, Rotavirus infection and Alzheimer’s disease, respectively [51,52,53,54,55].

It is possible that these probiotics could apply the same anti-inflammation properties to the ischemic stroke model which correlates with our included paper [14] using a mixture of probiotics that included *Bifidobacterium breve* and which showed anti-inflammation outcomes in an ischemic stroke model.

### 4.4. Probiotic and Oxidative Stress

Oxidative stress is a result of a disrupted redox pathway creating more reactive oxygen species (ROS) than antioxidants can reduce [56]. This phenomenon happens frequently in the neuron tissue due to its high energy production. Oxidative stress can cause cellular damage by nucleic acid damage, protein oxidation and lipids peroxidation [57].

The brain is composed of high lipid composition which causes it to be vulnerable to oxidative stress, especially in the lipid peroxidation pathway [58]. This can lead to multiple neurodegenerative disease and neuronal damage which could be prevented by the gut microbiota due to its ability to regulate endogenous and exogenous ROS. In this article our focus was on probiotics that have this potential, i.e., *Bifidobacterium breve*, *Lactobacillus casei*, *Lactobacillus bulgaricus* (L*actobacillus delbrueckii* subsp. *bulgaricus*), *Lactobacillus acidophilus* and inactivated lactobacillus [14,27]. This raises the importance of further study on probiotics that target this mechanism as a potential method to treat stroke patients in the future [59].

### 4.5. Low-Quality Paper with No Control or Vague Parameter

We found that many studies did not use case and control comparisons, which prevented us from being able to qualitatively interpret the results, and thus acts as a limitation of this review. Moreover, the variety of parameters used to evaluate the outcome of probiotic use made it difficult to form a conclusion from this review.

### 4.6. Further Research

This systematic review suggests that the lack of standardization among probiotic use in research poses a major obstacle in interpreting the role of probiotics in stroke treatment. Therefore, more in-depth studies of each probiotic are needed to consider changes to the mainstay of treatment in stroke patients. Moreover, the scarcity of in vivo studies further emphasizes the need to conduct more research on this topic. Investigating the effects of probiotic supplementation on stroke patients in greater detail could ultimately lead us to a cheaper and nonpharmacologic treatment option for post-stroke patients.

## 5. Conclusions

The current literature suggests that there are potential cognitive benefits of probiotic supplementation, especially among animal models, on decreasing cerebral volume, increasing neurological score, and decreasing the inflammatory response. However, more research is needed to confirm these suggestions in other populations.

## Figures and Tables

**Figure 1 nutrients-14-03661-f001:**
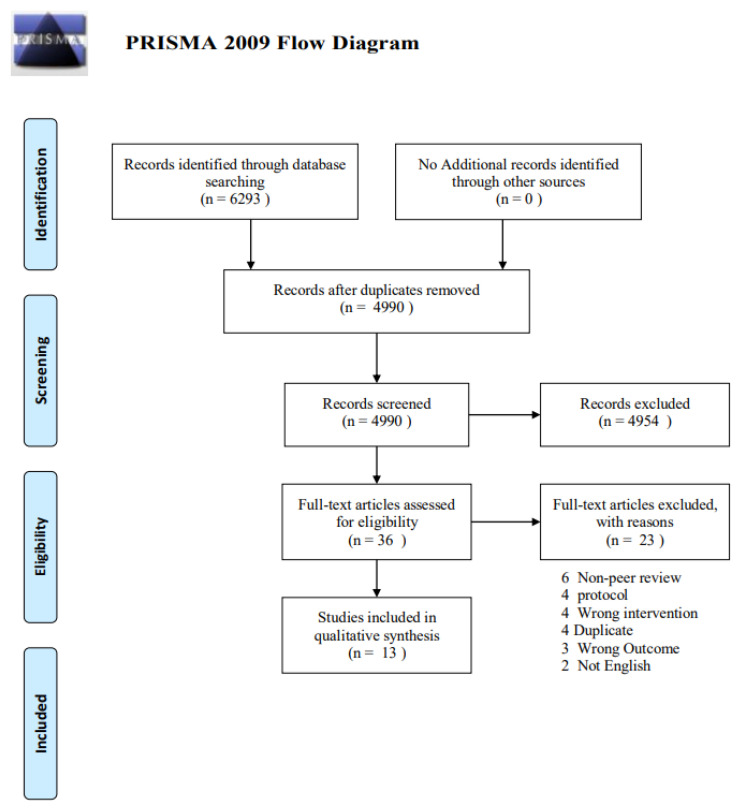
PRISMA flow diagram.

**Table 1 nutrients-14-03661-t001:** The characteristics of the articles in which probiotics are the intervention in stroke animal models.

Author	Country	Type of Intervention		Animal Type	Number of Case	Number of Control	Measure Outcome	Result of Outcome
		Non-food based	Food-based					
Akhoundzadeh, 2018 [14]	Iran	Combination of 4 viable probiotic bacteria strains, namely *Bifidobacterium breve*, *Lactobacillus casei*, *Lactobacillus bulgaricus* (*Lactobacillus delbrueckii* subsp. bulgaricus), and *Lactobacillus acidophilus*		Mice	5	5 mice received saline, 5 mice were sham operated	Neurological outcome	Could not improve neurological function
							Histopathology (infarct size)	Reduced infarct size 52%
							Biochemical markers	Decreased the malondialdehyde content and TNF alpha level
Bae, 2004 [17]	South Korea	Red ginseng and fermented red ginseng		Rat	Ginseng 5, red ginseng 5, fermented red ginseng 5	10	Histopathology (infarction area, volume)	Fermented red ginseng treated group showed reduction of the infarction area in all regions and total infarction volume
Chen R, 2019 [18]	United States		Puerariae Lobatae Radix(PLR) + Chuanxiong Rhizoma(CXR)	Rat	Not found	Not found	Neurological function score	Repaired neurological impairment
							Histopathology (body weight gain, cerebral infarction area)	Reduced the cerebral infarction
							Biochemical level (serum level of LDL/HDL/TG/T CHO/ blood viscosity/fibrinogen level/ platelet aggregation rate)	Reversed the dyslipidemiaReduced the blood viscosity and thrombotic risk
Li, 2018 [19]	United States		Panax Notoginsenoside extract(PNE)	Germ free rat	Not found	Not found	Neurological evaluation	Decreased neurological deficit scores
							Histopathology (triphenyl tetrazolium chloride (TTC) assessment of infarct size)	Decreased cerebral infarct volume
							Biochemical level (pro inflammatory cytokine/BDNF/GABA in rat hippocampus)	Upregulated the expression of GABA receptor in hippocampusDecreased rate of attenuation in BDNF expression
Liu, 2015 [20]	China	*Clostridium butyricum*		Mice	12	12	Neurological evaluation (behavioral tests, open field test, Morris water maze)	Improved spatial learning ability
							Histopathology (Hippocampal change)	Ameliorated the morphological changes in the HippocampusIncreased butyrate in the brain
Mei, 2017 [21]	China		Shuan tong ling	Rat	Not found	Not found	Neurological deficit	Increased neurological scores
							Histopathology (infarct volume)	Reduced infarct volume
							Biochemical level (inflammatory cytokines including TNF alpha, IL 1 beta)	Decreased TNF alpha and IL 1 beta
Nagao, 2019 [22]	Japan		Fermented ginseng	Rat	Not found	Not found	Neurological evaluation (spatial memory evaluated using Morris water maze (MWM))	Shortened the extended time to reach the platform in the MWM
							Histopathology (use of neuronal nuclei positive cells to assess hippocampus neuron loss, protein expression of caspase3/Iba1/glial fibrillary acidic protein)	Ameliorated loss of hippocampus cornu ammonis neurons and increased caspase-3/Iba1
Pang, 2020 [23]	United Kingdom		Yam gruel	Rat	9	18	Neurological evaluation (MWM test (spatial learning and memory function))	Improved cognitive function
							Biochemical markers (SOD and MDA, TNF alpha and IL 1 beta and LPS, characteristic of gut microbiota)	Increased relative content of probiotic bacteria and SCFAs in intestinal tract, cerebral cortexreduced oxidative stressand inflammatory responsePromoted the expression of neurotransmitters and brain derived neurotrophic factor
Park, 2016 [24]	Korea		Chungkookjang (fermented soybean)	Gerbil	Not found	Not found	Neurological evaluation	Prevented symptoms such as drooped eyelid/bristling hair/reduced muscle tone and flexor reflex/abnormal posture
							Histopathology (Neuronal cell death in hippocampus)	Prevent the neuronal cell deat
							Biochemical markers (cytokine expression in hippocampus, serum cytokine levels, glucose metabolism)	Suppress cytokine expression, prevent the impairment of glucose metabolism
Rahmati, 2019 [25]	Netherlands	Commercial probiotics (LactoCare capsule, 109 CFU, ZIST TAKHMIR, Tehran, Iran), which are a mixture of seven probiotic bacteria strains, including *Lactobacillus casei* ZT-Lca.106, *Lactobacillus acidophilus* ZT-Lac.51, *Lactobacillus rhamnosus* ZT-Lrh.54, *Lactobacillus bulgaricus* ZT-Lbu.90, Bifidobacterium breve ZT-Bbr.22, *Bifidobacterium longum* ZT-Lca.106, and *Streptococcus thermophilus* ZT-Sth.20		Mouse	30	20	Neurological evaluation (Spatial and learning memory)	Reduced spatial memory impairment and neurological dysfunction
							Histopathology (histological damage and apoptosis)	Reduced neuronal death
							Biochemical markers malondialdehyde (MDA) content and brain-derived neurotrophic factor (BDNF) level	MDA and BDNF change was not significant
Sun, 2016 [26]	Netherlands	Clostridium butyricum		Diabetic mice	Not found	Not found	Neurological evaluaiton (Cognitive impairment)	Ameliorate cognitive impairment
							Histopathology (neuronal injury, apoptosis, expression of Akt/p-Akt/caspase3 level)	Ameliorate histopathologic change in the hippocampusIncrease p-Atk expression and decreased caspase-3 expression equal inhibit neuronal apoptosis
							Biochemical markers (blood glucose level)	Decrease blood glucose level
Wanchao, 2018 [27]	China	Inactivated lactobacillus		Rat	24 (divided into 4 groups with varied concentrations)	6	Neurological evaluationneurolobehavioral scores,	Improve neurobehavioral scores
							HistopathologyCerebral infarction volume, tunnel and TLR4/ IkB/A20 (cell apoptosis)	Decrease cerebral infarction volumeDecrease neural cells apoptosisInhibit expression of TLR4Promote the expression of IkB and A20 which
							Biochemical markers (SOD + MDA levels),	MDA level decrease, SOD activity increase, reduce oxidative stress
Zhang, 2019 [28]	Netherlands		Chamomile	Rat	Not found	Not found	Neurological evaluation (neurological score, neurological deficits)	Improve neurological scores
							Histopathology (infarction size)	Decrease in both infarct volume
							Biochemical markers(Protein levels of Nrf2/Keap1/HO1/ERK)	Increase the activity of HO1 and Nrf2

## Data Availability

Not applicable.

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
