# Peer review of "Probiotic as a Potential Gut Microbiome Modifier for Stroke Treatment: A Systematic Scoping Review of In Vitro and In Vivo Studies"

_nutrients, 2022, doi:10.3390/nu14173661_

Round 1
Reviewer 1 Report
The manuscript, “Probiotic as a potential gut microbiome modifier for stroke treatment: A systematic scoping review of in vitro and in vivo studies.” Savigamin et al. reviewed the effect of probiotics on the stoke treatment and have included all the in vitro and in vivo evidence of probiotics on stoke up to May 7, 2021. The author included the data based on all the prebiotic, probiotic, and symbiotic studies. All the neurological tests, histological, and cytokines data have been described from the 13 selected manuscripts. However, the author has not included any in vitro studies in the review, and several edits are required.
Minor edits:
1. The author should improve the statement in the abstract “We searched in PubMed, Embase, and 16 Cochrane Central Register of Clinical up to May 7, 2021, and screened by two independent reviewers.”
2. It is not clear in the abstract what is the initial screening criteria.
3. The author should also improve the statement, “Histopathologically decreased loss of cerebral volume and inhibition of 22 neuronal apoptosis were found.” As histopathology is a technique, not an adverb.
4. Statement Line 35-36 “Several pharmacogenetic drugs have been investigated for use in stroke, including serotonergic drugs and dopaminergic drugs.” The reference cited by the author is not for pharmacogenetic drugs for serotonergic and dopaminergic drugs. Moreover, not all drugs are pharmacogenetic.
5. Authors should describe the reason for using these three databases, “PubMed, Embase, and Cochrane, and the criteria for choosing the independent reviewers.”
6. The author should change the word from Symbiotic to symbiotic throughout the manuscript.
7. The author should provide an abbreviation of most terms, such as “CSam, TN, IL-10, CA1, " throughout the manuscript.
8. Authors should describe what is considered wrong outcomes and interventions.
9. As author decided to review all the in vitro and in vivo studies included in the review. Then why does the author mention the statement in method line 119, “There was no in vitro study included.”
10. No comparison of studies was made based on the biochemical profile data other than cytokine.
11. The author should cite table 1 in the text. It would be better if table 1 should be divided into three different columns based on histological data, cytokine data, and the neurological test.
12. The statement “The gut microbiota-brain axis was also discussed in a study involving stroke and the microbiota, in which it was found that altering the gut microbiota could affect stroke outcome via bottom-up pathways.” Several studies describe the role of the stoke and gut axis and should be appropriately cited and improved.
13. The results should be described in detail.
Author Response
1. The author should improve the statement in the abstract “We searched in PubMed, Embase, and 16 Cochrane Central Register of Clinical up to May 7, 2021, and screened by two independent reviewers.”
Response: We changed to “A systematic review was conducted in PubMed, Embase, and Cochrane Central Register of Controlled Trials (CENTRAL) up to May 7, 2021. Titles and abstracts were screened and assessed by two independent reviewers. The initial screening criteria was aimed to include studies using probiotics, prebiotics, and symbiotics both in vitro and in vivo for the prevention and/or treatment of stroke”
2. It is not clear in the abstract what is the initial screening criteria.
Response: We added “The initial screening criteria was aimed to include studies using probiotics, prebiotics, and symbiotics both in vitro and in vivo for the prevention and/or treatment of stroke.”
3. The author should also improve the statement, “Histopathologically decreased loss of cerebral volume and inhibition of 22 neuronal apoptosis were found.” As histopathology is a technique, not an adverb.
Response: We changed to “Decreased loss of cerebral volume and inhibition of neuronal apoptosis was revealed in histopathological evidence.”
4. Statement Line 35-36 “Several pharmacogenetic drugs have been investigated for use in stroke, including serotonergic drugs and dopaminergic drugs.” The reference cited by the author is not for pharmacogenetic drugs for serotonergic and dopaminergic drugs. Moreover, not all drugs are pharmacogenetic.
Response: We changed to “Several monoaminergic drugs have been investigated for use in stroke, including serotonergic and dopaminergic drugs.”
5. Authors should describe the reason for using these three databases, “PubMed, Embase, and Cochrane, and the criteria for choosing the independent reviewers.”
Response: We added “We used three databases including PubMed, Embase, and Cochrane Central Register of Clinical Trials (CENTRAL) due to their coverage of international peer review articles.” and “The independent reviewers were selected for their experience in probiotics, stroke, and systematic review methodology.”
6. The author should change the word from Symbiotic to symbiotic throughout the manuscript.
Response: Symbiotic words were search and changed to symbiotic as advised.
7. The author should provide an abbreviation of most terms, such as “CSam, TN, IL-10, CA1, " throughout the manuscript.
Response: We added the List of Abbreviations section.
8. Authors should describe what is considered wrong outcomes and interventions.
Response: Wrong interventions include products other than probiotic, prebiotic, or symbiotic as an intervention. Wrong outcomes include non-neurological outcomes such as gastrointestinal symptoms.
9. As author decided to review all the in vitro and in vivo studies included in the review. Then why does the author mention the statement in method line 119, “There was no in vitro study included.”
Response: This review attempted to include both in vitro and in vivo studies but no in vitro studies passed the screening criteria used in this review.
10. No comparison of studies was made based on the biochemical profile data other than cytokine.
Response: We added the biochemical information as advised.
11. The author should cite table 1 in the text. It would be better if table 1 should be divided into three different columns based on histological data, cytokine data, and the neurological test.
Response: We divided the table into 3 sections with properly cited in the text as suggested.
12. The statement “The gut microbiota-brain axis was also discussed in a study involving stroke and the microbiota, in which it was found that altering the gut microbiota could affect stroke outcome via bottom-up pathways.” Several studies describe the role of the stoke and gut axis and should be appropriately cited and improved.
Response: We added “The gut microbiota-brain axis was also discussed in a study involving stroke and the microbiota, in which it was found that altering the gut microbiota could affect stroke outcome by impact both neuroendocrine and metabolic pathways including serotoninergic system, γ-Aminobutyric acid (GABA), plasma cortisol and catecholamine [8,9]. Moreover, previous data found that microbiota could impact central nervous system by using vagus nerve [10] and locally signal including histamine, acetylcholine, melatonin, nitric oxide and hydrogen oxide that could affect enteroendocrine cell. Microbiota had known to play an essential part of microglial maturation and activation which could change the behavior of the sample [11]. Furthermore, Mice that were given different food could result in different physical activity, memory and anxiety behavior which show how impact microbiota was via bottom-up pathways [9,12,13].“
13. The results should be described in detail.
Response: We added more biochemical and histopathological information.
Reviewer 2 Report
Dear authors,
your submitted manuscript requires stylistic and formal editing.
the following modifications are necessary:
- "in vitro" and "in vivo" must be in italic
- check dots and spaces throughout the text
- Line 134 - "de-crease" to "decrease"
- Line 189 - "neu-trophils" to "neutrophils"
Line 194 - "more-over" to "moreover"
and of course change all hyphenated words to right form
Discussion - this part is insufficient, I recommend adding more relevant findings and opinions from various studies.
Author Response
1. "in vitro" and "in vivo" must be in italic
Response: The two terms words were italicized.
2. check dots and spaces throughout the text
Response: The dots and spaces were checked and corrected.
3. Line 134 - "de-crease" to "decrease"
Response: Corrected.
4. Line 189 - "neu-trophils" to "neutrophils"
Response: Corrected.
5. Line 194 - "more-over" to "moreover" and of course change all hyphenated words to right form
Response: Corrected.
6. Discussion - this part is insufficient; I recommend adding more relevant findings and opinions from various studies.
Response: We have expanded the discussion as advised.
Round 2
Reviewer 2 Report
Thr submitted manuscript was improved.